# Comparative Blood Transcriptome Analysis of Semi-Natural and Controlled Environment Populations of Yangtze Finless Porpoise

**DOI:** 10.3390/ani14020199

**Published:** 2024-01-07

**Authors:** Wang Liu, Denghua Yin, Zhanwei Li, Xiaoyan Zhu, Sigang Zhang, Peng Zhang, Danqing Lin, Zhong Hua, Zhichen Cao, Han Zhang, Jialu Zhang, Congping Ying, Pao Xu, Guixin Dong, Kai Liu

**Affiliations:** 1Key Laboratory of Freshwater Fisheries and Germplasm Resources Utilization, Ministry of Agriculture and Rural Affairs, Freshwater Fisheries Research Center, Chinese Academy of Fishery Sciences, Wuxi 214081, China; liuwang1357@outlook.com (W.L.); yindenghua@ffrc.cn (D.Y.); lindq@ffrc.cn (D.L.); huazhong@ffrc.cn (Z.H.); zhangjialu@ffrc.cn (J.Z.); xup@ffrc.cn (P.X.); 2Wuxi Fisheries College, Nanjing Agricultural University, Wuxi 214081, China; 2020213002@stu.njau.edu.cn; 3Zhuhai Chimelong Investment & Development Co., Ltd., Zhuhai 519000, China; lizw@chimelong.com (Z.L.); caesar@chimelong.com (P.Z.); 4Anqing Aquatic Technology Promotion Center Station, Anqing 246000, China; aqyyjsck@163.com (X.Z.); zsg910@163.com (S.Z.); 5National Demonstration Center for Experimental Fisheries Science Education, Shanghai Ocean University, Shanghai 201306, China; czcdr007@163.com (Z.C.); m220100406@st.shou.edu.cn (H.Z.); 6Guangdong South China Rare Wild Animal Species Conservation Center, Zhuhai 519031, China

**Keywords:** Yangtze finless porpoise, blood transcriptome, environment, energy metabolism, auditory

## Abstract

**Simple Summary:**

The Yangtze finless porpoise (*Neophocaena asiaeorientalis asiaeorientalis*) is an endangered freshwater whale unique to the Yangtze River in China. We compared and analyzed the gene expression of the Yangtze finless porpoise in semi-natural and controlled environments, based on blood transcriptomics methods. The results showed that the acoustic behavior and auditory-related genes of the Yangtze finless porpoise may show responsive changes and differential expression under different environment conditions, and thus the auditory sensitivity may also show corresponding adaptive characteristics. This study provides a new perspective for the further exploration of the responsive changes of the two populations to various environments and provides a theoretical reference for further improvements in conservation practices for the Yangtze finless porpoise.

**Abstract:**

The Yangtze finless porpoises (*Neophocaena asiaeorientalis asiaeorientalis*) living in different environments display significant differences in behavior and physiology. To compare and analyze gene expression differences between an ex situ population and a controlled environment population of the Yangtze finless porpoise, we sequenced the transcriptome of blood tissues living in a semi-natural reserve and an artificial facility, respectively. We identified 6860 differentially expressed genes (DEGs), of which 6603 were up-regulated and 257 were down-regulated in the controlled environment vs ex situ comparison. GO and KEGG enrichment analysis showed that the up-regulated genes in the controlled environment population were significantly associated with glucose metabolism, amino acid metabolism, and the nervous system, while those up-regulated in the ex situ population were significantly associated with energy supply and biosynthesis. Further analysis showed that metabolic and hearing-related genes were significantly affected by changes in the environment, and key metabolic genes such as *HK*, *PFK*, *IDH*, and *GLS* and key hearing-related genes such as *OTOA*, *OTOF*, *SLC38A1*, and *GABBR2* were identified. These results suggest that the controlled environment population may have enhanced glucose metabolic ability via activation of glycolysis/gluconeogenesis, the TCA cycle, and inositol phosphate metabolism, while the ex situ population may meet higher energy requirements via enhancement of the amino acid metabolism of the liver and muscle and oxidative phosphorylation. Additionally, the acoustic behavior and auditory-related genes of Yangtze finless porpoise may show responsive changes and differential expression under different environment conditions, and thus the auditory sensitivity may also show corresponding adaptive characteristics. This study provides a new perspective for further exploration of the responsive changes of the two populations to various environments and provides a theoretical reference for further improvements in conservation practices for the Yangtze finless porpoise.

## 1. Introduction

The Yangtze finless porpoise (*Neophocaena asiaeorientalis asiaeorientalis*) is a small cetacean endemic to China, which is mainly distributed in the middle and lower reaches of the Yangtze River, Dongting Lake and Poyang Lake, and tributary tailbacks. Due to the continuous deterioration of the ecological environment of the Yangtze River in recent years, the future population status of the Yangtze finless porpoise is of great concern. Scientific surveys of the Yangtze finless porpoise in 2006, 2012, and 2017 found population numbers of about 1800, 1040 and 1012 [1,2], showing a downward trend year by year. The IUCN Red List of Threatened Species categorized the Yangtze finless porpoise as “critically endangered” in 2013. In 2022, the most recent scientific survey found that the population of Yangtze finless porpoise had risen to 1249, an increase of 23.42% over 2017. Although the downward trend of the Yangtze finless porpoise population has seemingly been reversed, in view of the long cycle of environmental restoration in the Yangtze River and the realistic needs of economic development, the living conditions of the Yangtze finless porpoise are still poor, and the road to protect the Yangtze finless porpoise still has a long way to go.

Ex situ conservation and artificial breeding are two important approaches to protect the Yangtze finless porpoise. The ex situ conservation area is a semi-natural environment, while the artificial breeding base is a fully artificial environment. Studies have shown that Yangtze finless porpoises living in different environments exhibit significant differences in behavior and physiology. In comparisons of Yangtze finless porpoise populations in Poyang Lake (a natural environment) and the Tian-E-Zhou Oxbow reserve (a semi-natural environment), concentrations of serum thyroid hormone and cortisol hormone were found to be significantly higher in the former than in the latter, while the concentrations of neutrophils and platelets were found to be significantly higher in the latter [3,4]. Lin et al. compared the serum amino acid content of the natural and controlled environment populations of the Yangtze finless porpoise and found that the contents of 14 amino acids were significantly higher in the controlled environment population than the natural population. In addition, acoustic experiments revealed that Yangtze finless porpoises living in different environments have their own acoustic behaviors and acoustic signal characteristics [5]. Together, these studies convincingly show that the phenotypic traits of the Yangtze finless porpoise are dependent on the environment. However, due to the particularity of the research subjects and the restriction of experimental conditions, there is still a lack of comparative studies between semi-natural populations and controlled environment populations, especially the verification at the molecular level, and the molecular mechanisms underlying the environmental adaptability of the Yangtze finless porpoise remain unclear.

Transcriptome sequencing can capture the expression of genes in real time [6]. By detecting the expression levels of genes in specific tissues, genes and metabolic pathways related to specific traits can be identified so as to better understand the adaptive mechanisms of organisms to environmental changes. For rare wild animals such as the Yangtze finless porpoise, blood is particularly feasible to use for transcriptome research. First, blood is easier to obtain than other tissues like organs and muscles, and blood sampling results in minimal harm to the experimental animals [7]. Second, blood flows through all parts of the body via the circulatory system, connecting all tissues and organs as a whole, and thus is useful for studying various physiological processes, including the immune response [8], energy metabolism, and stress response [9]. 

To understand gene expression differences of the Yangtze finless porpoise in ex situ and controlled environments and to investigate the impact of habitat types on the Yangtze finless porpoise and its molecular regulation mechanisms, we used RNA-Seq to sequence and compare the transcriptomes of blood tissues from Yangtze finless porpoises living in an ex situ reserve and an artificial facility, respectively. Identification of the genes and functional pathways most significantly affected by environment will better elucidate the key genes and pathways underlying environmental adaptability. This study will serve as a starting point for further investigation of the adaptability of Yangtze finless porpoises to diverse environments and provide theoretical support to improve the quality of Yangtze finless porpoise environment.

## 2. Materials and Methods

### 2.1. Animals and Sample Collection

In this study, six blood samples were collected from healthy Yangtze finless porpoises, including three YFPs with an average age of 5 years old from the Anqing Xijiang Yangtze finless porpoise ex situ conservation base (XJ_1♂, XJ_2♂, XJ_3♂) in March 2018 and three YFPs with an average age of 7 years old from the Zhuhai Chimelong Yangtze finless porpoise artificial breeding and science popularization education base (CA_1♂, CA_2♀, CA_3♂) in November 2018. See Appendix A for specific information. As an oxbow running parallel to the Yangtze River, the Anqing Xijiang Yangtze finless porpoise ex situ conservation base was formally established in 2016. The total length of the reserve is about 9 km, the average water depth is about 8.7 m, and the maximum water depth exceeds 20 m. Both sides of the reserve are natural shorelines, with an excellent ecological environment and management conditions. The Yangtze finless porpoise artificial breeding pool is equipped with an independent water life-support system, and the water temperature is constant at about 25 °C all year round. Samples were divided into two groups according to environment types, namely the ex situ conservation water group and the artificial breeding group. Blood samples from six Yangtze finless porpoises were collected during routine physical examinations. Each blood sample was extracted from the main tail vein and then transferred to a vacuum collecting vessel (PAXgene^TM^, Franklin Lakes, NJ, USA). A total of 2 mL blood samples were then transported back to the laboratory and stored in the −80 °C ultra-low temperature refrigerator for future use.

### 2.2. Total RNA Extraction and cDNA Library Preparation

Total RNA was extracted from whole blood samples from six Yangtze finless porpoises using the PAXgene Blood RNA Kit (QIAGEN, Hilden, Germany) according to manufacturer’s instructions, and the quality of RNA was checked using an Agilent 2100 Bioanalyzer (Agilent, Santa Clara, CA, USA). High-quality RNA was used for subsequent library construction. The cDNA library was constructed using the MGIEasy RNA library preparation kit (Huada, Wuhan, China). Oligo(dT)-attached magnetic beads were used to enrich mRNA, and a fragmentation buffer was used to fragment the obtained mRNA. cDNA was synthesized using the resulting fragments as templates. After incubation, A-tailing mixture and RNA index adapters were added for terminal repair. Suitable fragments were selected as the templates for PCR amplification using agarose gel electrophoresis. PCR products were thermally denatured into single-stranded DNA, and a single-stranded DNA library was obtained by cyclizing the single-stranded DNA with a bridge primer. 

### 2.3. mRNA Sequencing and Analysis

After libraries were successfully constructed, high-throughput sequencing was performed using the MGI-2000 platform (Huada, Wuhan, China). The raw data generated by sequencing were filtered using SOAPnuke (v1.4.0) software [10] to remove reads with sequencing adapters, ‘N’ base ratios over 5%, and low-quality base ratios over 20% to obtain clean reads. The resulting clean reads were aligned to the Yangtze finless porpoise reference genome using HISAT2 (v1.4.0) software [11], and then the gene expression levels for each sample were calculated using RSEM software (v1.2.8) [12]. Gene expression levels were expressed as FPKM (fragments per kilobase of transcript per million mapped fragments).

DEseq2 (v1.4.5) [13] was used for the identification of differentially expressed genes (DEGs), with DEGs defines as genes with |log2foldchange| ≥ 1 and q ≤ 0.001 (adjusted *p* value). Functional classification and biological pathway classification of DEGs were performed using the GO (Gene Ontology) and KEGG (Kyoto Encyclopedia of Genes and Genomes) databases, while enrichment analysis was performed using a hypergeometric test via the phyper function.

### 2.4. Real-Time Quantitative PCR Validation

To verify the accuracy of transcriptome sequencing results, eight differentially expressed genes were randomly selected for real-time quantitative PCR. *GAPDH* (glyceraldehyde 3-phosphate dehydrogenase) was selected as the endogenous control gene. Primer sequences (Appendix A) were designed using Primer Premier 5 (Premier Biosoft 5.0, San Francisco, CA, USA). The HiScript III RT SuperMix for qPCR kit (Vazyme, Nanjin, China) was used for reverse transcription of mRNA. All qRT-PCR was performed using the ChamQ Universal SYBR qPCR Master Mix (Vazyme, China). The reaction was carried out with a total volume of 20 μL, including 10 μL qPCR Mix, 0.8 μL of primers (10 μM), 2 μL of cDNA (500 ng), and 7.2 μL ddH_2_O. The following PCR program was applied in two steps: denaturation at 95 °C for 30 s followed by 40 cycles consisting of 95 °C for 10 s and 60 °C for 30 s. qRT-PCR was performed using a StepOne Software v2.0 fluorescent quantitative PCR instrument (ABI, Foster, CA, USA), and relative gene expression was calculated using the 2^−ΔΔCT^ method. 

### 2.5. Supporting Information

mRNA clean transcriptome data of CA_1, CA_2 and CA_3 have been uploaded to the Sequence Read Archive (SRA) under accession number PRJNA980390. mRNA clean transcriptome data of XJ_1, XJ_2 and XJ_3 have been uploaded to SRA under accession number PRJNA789349 [14].

## 3. Results

### 3.1. Sequencing Data and Alignment Analysis

The transcriptome libraries of the Yangtze finless porpoise blood samples were sequenced using the MGI-2000 platform, and a total of 514,060,000 raw reads were obtained. After removing low-quality reads, reads containing sequencing adapters, and reads with excessive unknown base ‘N’ contents, a total of 501,970,000 clean reads were obtained. The average size of each sample was 8.37 Gb, and the six samples had base quality values of Q20 ≥ 97% and Q30 ≥ 93%. Clean reads from each sample were aligned to the reference genome, with an average alignment rate of 79.86%. The average alignment rate of uniquely mapped reads was 48.03% (Table 1). The transcriptome sequencing quality was satisfactory, meaning the constructed libraries could be used for subsequent gene expression analysis.

### 3.2. Analysis and Functional Annotation of Differentially Expressed Genes

Differential expression analysis of the above high-quality transcriptome data identified a total of 6860 DEGs between the XJ and CA groups, of which 6603 were significantly up-regulated in the CA group and 257 were significantly down-regulated in the XJ group (Appendix A), indicating that environment changes had a significant impact on the gene expression profiles of the Yangtze finless porpoise.

To further understand the physiological functions regulated by these DEGs, functional annotation of DEGs was performed using GO and KEGG databases. Using the GO database, 5050 DEGs were annotated with GO categories, including “biological process” (3004), “cellular component” (3276), and “molecular function” (3995) (Appendix A). A total of 3290 DEGs were annotated using the KEGG database, including “environmental information processing” (1153), “genetic information processing” (767), “cellular processes” (1078), “organismal systems” (1373), and “metabolism” (815) (Appendix A). Further analysis revealed that the 815 DEGs annotated with “metabolism” were highly enriched for 10 common metabolic pathways (Table 2). In addition, 150 DEGs had annotations for environmental adaptation pathways. These were mainly involved in the regulation of the activities of the nervous system, endocrine system, and sensory system, indicating that the Yangtze finless porpoise may adapt to different environment conditions through physiological regulation of the above aspects. Table 2 shows the top 10 most significantly enriched KEGG pathways among these DEGs.

### 3.3. Enrichment Analysis of Differentially Expressed Genes

GO enrichment analysis of DEGs showed that the 6860 DEGs were significantly enriched for 785 GO terms and extremely significantly enriched for 140 GO terms, among which “nucleoplasm” and “intracellular” were the most significant. The up-regulated genes of the CA group were significantly enriched for GO terms including “nucleoplasm”, “intracellular membrane-bounded organelle”, “transferase activity”, and “regulation of primary metabolic process” (Figure 1A). The up-regulated genes of the XJ group were significantly enriched for GO terms including “structural constitution of ribosome”, “ribosome”, “peptide biosynthetic process”, and “organonitrogen compound biosynthetic process” (Figure 1B). The up-regulated genes of Yangtze finless porpoise in different environments have obvious differences in biological function.

KEGG enrichment analysis of DEGs showed that the 6860 DEGs were significantly enriched for 58 KEGG pathways and extremely significantly enriched for 12 KEGG pathways. Figure 2A shows the top 30 KEGG pathways with the most significant enrichment. The up-regulated genes of the CA group were significantly enriched for KEGG pathways such as “inositol phosphate metabolism”, “N-Glycan biosynthesis”, “lysine degradation”, and “inflammatory mediator regulation of TRP channels” (Figure 1C), which were mainly involved in the regulation of glucose metabolism, amino acid metabolism, and sensory physiology. The up-regulated genes of the XJ group were significantly enriched for “ribosome”, “hematopoietic cell lineage”, “oxidative phosphorylation”, “porphyrin and chlorophyll metabolism”, and “hippo signaling pathway” (Figure 1D).

### 3.4. Analysis of Key Genes

In this study, we found that expression levels of metabolic and hearing-related genes were significantly affected by the environment. For example, DEGs such as *INPP4A*, *INPP5A*, *IMPA1*, and *MIOX* were significantly up-regulated in the CA group, suggesting the activation of the inositol phosphate metabolism pathway and thus the acceleration of glucose absorption in the blood. *GNAO1*, *GABBR2*, *IQC1*, *PLD1*, and other genes in the glutamatergic synapse and GABAergic synapse pathway were significantly up-regulated in the CA group, regulating neurotransmitter transmission in the auditory system (Figure 3). In addition, we also found that several metabolic genes (*HK*, *PFK*, *CS*, and *ATP5PD*) and hearing-related genes (*MYO6*, *MYO7A*, *OTOA*, and *OTOF*) were significantly differentially expressed between the CA group and the XJ group (Table 3). Combined with existing literature, we speculate that these genes may play an important role in the adaptation of the Yangtze finless porpoise to diverse environments such as ex situ reserve and artificial environments. According to the cluster heat map (Figure 2B), the expression of *NDUFA1*, *NDUFA6*, *NDUF5S*, and *ATP5PD* genes in the oxidative phosphorylation pathway were significantly up-regulated in the XJ group, while most other genes were up-regulated in the CA group, indicating that environment changes had a significant impact on the metabolism and hearing system of the Yangtze finless porpoise.

### 3.5. Validation of RNA-Seq Results by qRT-PCR

The results of qRT-PCR validation of eight randomly selected differentially expressed genes are shown in Figure 4. IMPA1, INPP5A, HK2, SLC38A1, and INPP5B were up-regulated in the CA group, while NDUFA6, ATP5PD, and ITGA2B were down-regulated in the CA group compared with the XJ group. The differential gene expression verified by qRT-PCR was generally consistent with the transcriptome sequencing results, indicating that the transcriptome sequencing results were reliable in this study.

## 4. Discussion

### 4.1. Differences in Carbohydrate and Amino Acid Metabolism in Different Environments of the Yangtze Finless Porpoise

In this study, we found that a number of metabolic genes were significantly differentially expressed between Yangtze finless porpoise populations living in different environments, and these DEGs were significantly enriched for “inositol phosphate metabolism”, “lysine degradation”, “N-glycan biosynthesis”, “amino sugar and nucleotide sugar metabolism”, and other pathways that regulate carbohydrate and amino acid metabolism. Under normal circumstances, the body’s blood glucose concentration is relatively stable; that is, the source and destination of blood glucose maintain a dynamic balance [15]. In this study, both glycogen synthesis and glycogen decomposition genes were preferentially expressed in the controlled environment population, indicating enhanced carbohydrate metabolism in the controlled environment population as compared to the ex situ population. Research has shown that inositol phosphate can reduce insulin resistance in organisms and increase insulin sensitivity, thus accelerating glucose absorption in the blood [16]. The activation of the inositol phosphate metabolism pathway in the controlled environment population indicates that the glucose metabolic rates of Yangtze finless porpoises in this environment were faster. In addition, the rate-limiting enzymes hexokinase and phosphofructokinase in the glycolysis pathway were also up-regulated in the controlled environment population. Citrate synthase, isocitrate dehydrogenase, and aconitate hydratase from the TCA pathway were also up-regulated in the controlled environment population, further suggesting that the glucose metabolic activities of the controlled environment population were higher than those of the ex situ population.

A previous study on the energy intake of Yangtze finless porpoises showed that seasonal changes in water temperature led to significant seasonal changes in energy intake, with the lowest energy intake rates in spring and the highest in winter. However, when the water temperature was maintained at 17–26 °C, the total monthly food intake of Yangtze finless porpoises remained stable. In this study, the sampling time for the Yangtze finless porpoises in the ex situ reserve was in spring (March), while the water in the artificial breeding pool was maintained at a constant temperature of 25 °C, indicating that there were differences in energy intake between finless porpoises living in the ex situ reserve and artificial facility. Due to the influence of water temperature and seasonal factors, the energy intake rates of finless porpoises in the ex situ reserve are low in spring, and thus their metabolic levels would be lower than other seasons. The controlled environment population, on the other hand, has a stable and high-quality food source, so their daily glucose metabolic processes would be more active than those of the ex situ population.

In addition to glucose metabolism, amino acid metabolism is also an important component of energy metabolism in organisms. Lin et al. compared the serum amino acid content of the natural and controlled environment populations of the Yangtze finless porpoise and found that amino acid contents, except proline, methionine, and histidine, were significantly or extremely significantly higher in the controlled environment population than the natural population. The study pointed to two main reasons for the differences in amino acid contents. First, the metabolic rates of finless porpoises living in open waters would be increased due to the strong demand for foraging and exercise, and glutamic oxaloacetic transaminase and glutamic pyruvic transaminase activities in animals with high metabolic rates would also be increased [17]. Thus, the liver’s function of transforming amino acids would be strengthened, and a large number of amino acids in the blood would enter the liver for metabolism [18]. Second, finless porpoises in natural habitats have a wide range of activities and high energy consumption demands, and as various proteins are the material basis of muscle contraction [19], the natural finless porpoises would need to take up more amino acids from the blood for muscle metabolism. Consistent with this, this study found that amino acid metabolism-related genes were significantly up-regulated in the blood tissues of the controlled environment population, indicating that the controlled environment population had less muscle exercise and lower energy consumption demands. In addition, this study also found that the up-regulated genes of finless porpoises in the ex situ reserve, including *NDUFA1*, *NDUFA6*, *NDUF5S*, and *ATP5PD*, were significantly enriched for the oxidative phosphorylation pathway. These genes encode important catalytic enzymes in oxidative phosphorylation processes and drive ATP synthesis [20]. Thus, the ex situ population may have higher energy demands than the controlled environment population.

### 4.2. Hearing Differences of Yangtze Finless Porpoises in Different Environments

Previous studies have shown that there are significant differences in immunity [21], metabolism [22], and stress [4] among Yangtze finless porpoise species from different environments. However, little attention has been paid to the effects of environment changes on the sensory physiology of the Yangtze finless porpoise. As a special echolocation species, the Yangtze finless porpoise mainly relies on high-frequency sound waves emitted and reflected back to itself for positioning, predation, environmental detection, and other basic life activities and thus possesses a highly developed high-frequency hearing ability [23]. However, in the artificial facility, the echolocation behavior of the Yangtze finless porpoise may change adaptively due to the limited space. Studies have shown that *MYO6*, *MYO7A*, *OTOF*, *OTOA*, and *TECTB* are potential echolocation genes that are crucial for high-frequency hearing production in mammals [24,25,26]. *MYO6* and *MYO7A* belong to the myosin family and are responsible for auditory hair cell transduction in mammals [26]. *OTOF* encodes otoferlin and is closely related to the synaptic transmission of auditory neurons [27]. The *OTOA*-encoded protein is a non-cytoglial component specific to the inner ear, and hearing impairment has been observed in mutants [24]. *TECTB* mainly acts on the mechanical conduction of sound [28]. In this study, we found that expression levels of the above genes in the blood tissues of the Yangtze finless porpoise were generally low, though this may be related to the fact that they mainly play roles in hearing. However, their expression levels in the controlled environment population were still significantly higher than the ex situ population, suggesting that different environment conditions may lead to adaptive changes in the echolocation behavior of Yangtze finless porpoises. Normally, water space in captivity is limited, so the Yangtze finless porpoises living therein need to shorten acoustic signal transmission intervals to achieve continuous positioning and avoid collisions with the pool wall. Previous studies on the acoustic signals of Yangtze finless porpoises have also confirmed this point. The echolocation signals of Yangtze finless porpoises in natural waters increased significantly when they were put into a seine net. After a period of captivity, echolocation signals decreased, and such signals increased significantly only during detection tasks such as predation, which is speculated to be due to the finless porpoise’s increasing familiarity with the artificial environment [29].

In mammals, glutaminergic synapses and GABAergic synapses are both important synapses regulating neural excitation in the auditory system, among which glutamic acid is the main excitatory neurotransmitter [30] and GABA is the main inhibitory neurotransmitter [31]. In this study, *SLC38A1A*, *GABBR2*, *GNAO1*, and other differentially expressed genes were significantly enriched for glutamatergic synapse and GABAergic synapse pathways. These may affect the auditory sensitivity of Yangtze finless porpoises living in different environments. Studies have shown that during the transmission of acoustic signals to the neural centers of the brain, presynaptic cells can achieve rapid excitatory inhibition through the release of GABA to balance neural excitation generated by glutamate and ultimately reduce the sensitivity of the auditory response [32]. Lu et al. [33] conducted transcriptome studies using the inner ear of turtles and found that male turtles’ auditory sensitivity was weaker than that of female turtles, possibly because multiple genes involved in the GABAergic synaptic pathway were significantly up-regulated in male turtles, and thus activation of this pathway slowed down the auditory response speed of the nerve center. Yang et al. [34] identified several highly expressed hearing-related genes in the swim bladder tissue of channel catfish, which were significantly enriched for the glutamatergic synaptic pathway and GABAergic synaptic pathway. All of the above studies indicated that the glutaminergic synaptic pathway and GABAergic synaptic pathway were closely associated with hearing function. In this study, the genes involved in these two pathways were overexpressed in the controlled environment population, suggesting that the auditory sensitivity of the Yangtze finless porpoise under different environment conditions may show environmental adaptation characteristics.

## 5. Conclusions

In this study, transcriptomic analysis of blood tissues collected from Yangtze finless porpoises living in a semi-natural and an artificial facility, respectively, was performed using RNA-Seq technology. A total of 6860 DEGs were identified, of which 6603 were up-regulated in the controlled environment population and 257 were up-regulated in the ex situ population. Further analysis showed that metabolic and hearing-related genes were significantly affected by environment changes. Key metabolic genes such as *HK*, *PFK*, *IDH*, and *GLS* and key hearing-related genes such as *OTOA*, *OTOF*, *SLC38A1*, and *GABBR2* were identified as DEGs from the two populations. These results suggest that the controlled environment population may have enhanced glucose metabolic ability via activation of glycolysis/gluconeogenesis, the TCA cycle, and inositol phosphate metabolism, while the ex situ population may meet higher energy requirements via enhancement of the amino acid metabolism of liver and muscle and oxidative phosphorylation. Additionally, the acoustic behavior and auditory-related genes of Yangtze finless porpoise may show responsive changes and differential expression under different environment conditions, and thus the auditory sensitivity may also show corresponding adaptive characteristics. Our research team conducted a similar study at the same time. We performed RNA sequencing of blood tissues from Yangtze finless porpoises in ex situ and controlled environments, focusing on the impacts of environmental changes on genes related to vision, digestion, and immune system. It was found that the ex situ population may enhance their immune function to adapt to the complex semi-natural environment, while the visual and protein digestion functions of the population in the controlled environment were improved [35]. In this study, we mainly focused on the expression of metabolic and hearing-related genes between an ex situ population and a controlled population of the Yangtze finless porpoise.

In conclusion, the two different environmental conditions of ex situ conservation and artificial breeding will affect the expression of blood genes in Yangtze finless porpoises. Therefore, the conservation of YFP requires the selection of appropriate methods according to the specific situation and the consideration of various factors in order to achieve the best results. In order to minimize the impact of the physiological condition of the Yangtze finless porpoise on the experimental results, we selected six physically healthy adult Yangtze finless porpoises with similar ages. We mainly focused on the between-group differences of Yangtze finless porpoises in the two environments rather than the within-group differences due to different sexes or ages. This study preliminarily revealed differences in gene expression of the blood transcriptomes of Yangtze finless porpoises living in different environments and initially explored the multiple effects of environment factors on the Yangtze finless porpoise, providing a theoretical reference for the further improvement of conservation practices to protect this endangered aquatic mammal.

## Figures and Tables

**Figure 1 animals-14-00199-f001:**
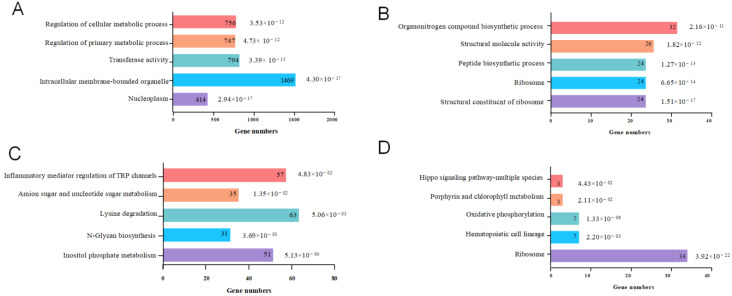
GO and KEGG enrichment of DEGs. (**A**) GO enrichment of up-regulated DEGs in CA; (**B**) GO enrichment of up-regulated DEGs in XJ; (**C**) KEGG enrichment of up-regulated DEGs in CA; (**D**) KEGG enrichment of up-regulated DEGs in XJ. The numbers inside the bars indicate enriched genes. Numbers outside the bars are *p*-values.

**Figure 2 animals-14-00199-f002:**
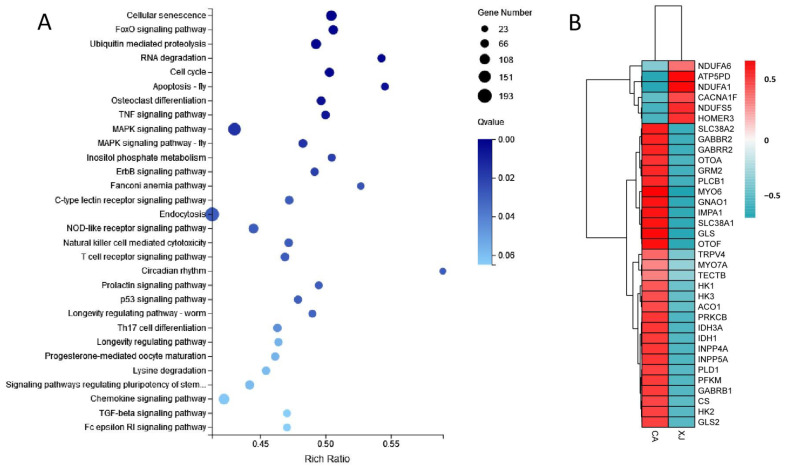
KEGG enrichment analysis of DEGs and heatmap of key genes. (**A**) Top 30 significantly enriched KEGG pathways. (**B**) The heatmap of key genes. Red indicates up-regulation, blue indicates down-regulation, and the shade of color indicates the degree of regulation.

**Figure 3 animals-14-00199-f003:**
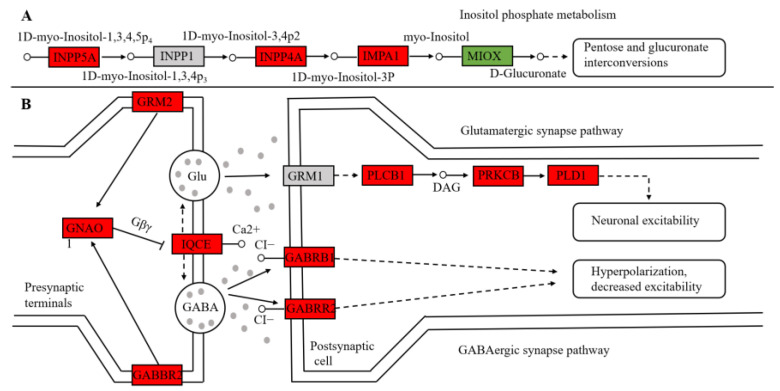
Major signaling pathways in the blood tissue of the Yangtze finless porpoise. (**A**) Inositol phosphate metabolism pathway; (**B**) glutamatergic and GABAergic synapse pathway. Red represents up-regulation in CA while green represents down-regulation in CA.

**Figure 4 animals-14-00199-f004:**
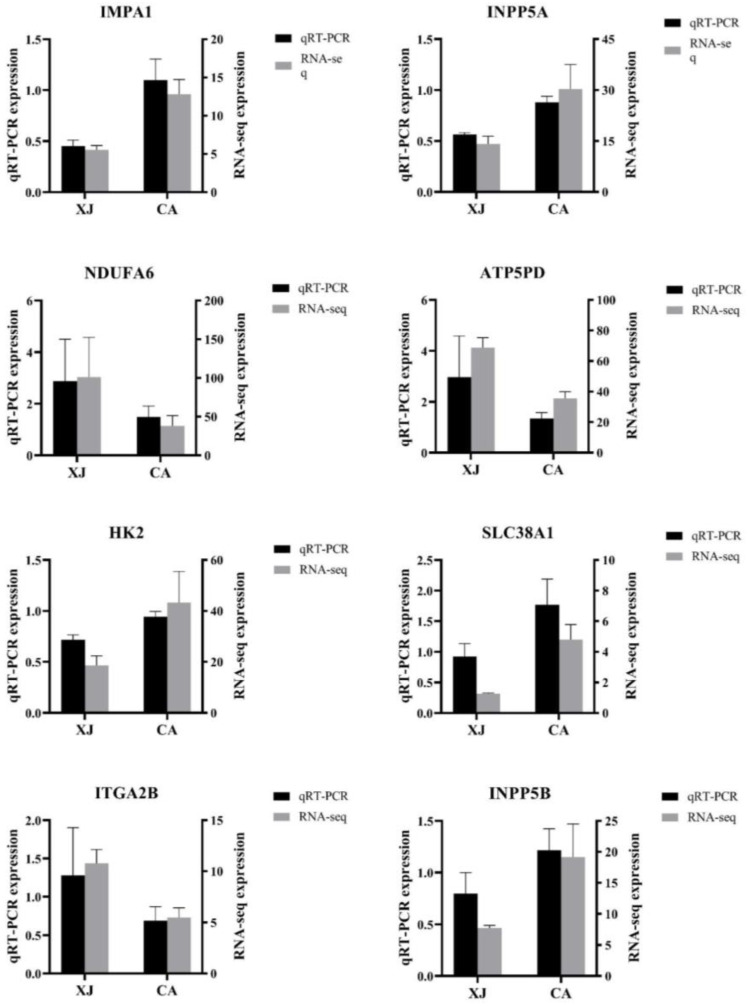
Quantitative real-time PCR for validation of RNA-Seq. qRT-PCR verification of eight differentially expressed genes in CA vs XJ groups. The left vertical axis represents qRT-PCR relative expression. The right vertical axis represents RNA-Seq expression.

**Table 1 animals-14-00199-t001:** Sequencing and mapping statistics of blood samples.

Sample	Total Raw Reads (M)	Total Clean Reads (M)	Total Clean Bases (Gb)	Clean Reads Q20 (%)	Clean Reads Q30 (%)	Total Mapping (%)	Uniquely Mapping (%)
CA_1	128.75	126.62	12.66	97.98	93.98	89.39	68.60
CA_2	75.36	73.54	7.35	97.94	93.85	75.64	39.80
CA_3	88.15	86.37	8.64	98.00	94.06	84.83	60.10
XJ_1	66.32	64.62	6.46	98.02	94.09	77.62	41.93
XJ_2	69.75	67.55	6.76	98.09	94.28	73.90	34.36
XJ_3	85.73	83.27	8.33	98.09	94.28	77.75	43.38

**Table 2 animals-14-00199-t002:** Significantly enriched pathways of Yangtze finless porpoise DEGs related to metabolism and environmental adaptation.

Pathway ID	Pathway Name	Functional Classification	*p* Value
Significantly enriched pathways of DEGs related to metabolism			
ko00562	inositol phosphate metabolism	carbohydrate metabolism	1.42 × 10^−28^
ko00520	amino sugar and nucleotide sugar metabolism		5.78 × 10^−18^
ko00010	glycolysis/gluconeogenesis		3.32 × 10^−12^
ko00020	citrate cycle (TCA cycle)		1.12 × 10^−10^
ko00620	pyruvate metabolism		1.37 × 10^−10^
ko00510	N-glycan biosynthesis	glycan biosynthesis and metabolism	1.28 × 10^−17^
ko00330	arginine and proline metabolism	amino acid metabolism	1.19 × 10^−14^
ko00270	cysteine and methionine metabolism		1.28 × 10^−12^
ko00563	valine, leucine and isoleucine degradation		1.84 × 10^−9^
ko00471	D-glutamine and D-glutamate metabolism		1.19 × 10^−14^
Significantly enriched pathways of DEGs related to environmental adaptation			
ko04714	thermogenesis	environmental adaptation	6.53 × 10^−106^
ko04912	GnRH signaling pathway	endocrine system	4.31 × 10^−25^
ko04921	oxytocin signaling pathway		4.67 × 10^−34^
ko04725	cholinergic synapse	nervous system	7.43 × 10^−31^
ko04728	dopaminergic synapse		1.06 × 10^−29^
ko04723	retrograde endocannabinoid signaling		3.00 × 10^−28^
ko04724	glutamatergic synapse		3.96 × 10^−21^
ko04727	GABAergic synapse		3.72 × 10^−13^
ko04750	inflammatory mediator regulation of TRP channels	sensory system	1.16 × 10^−19^
ko04745	phototransduction-fly		1.84 × 10^−13^

**Table 3 animals-14-00199-t003:** Expression of key DEGs in the Yangtze finless porpoise.

Gene Name	Description	Gene ID	log_2_FC (CA/XJ)
metabolism-related differentially expressed genes			
HK1	hexokinase 1	112413526	1.03
HK2	hexokinase 2	112403280	1.33
HK3	hexokinase 3	112397869	1.11
PFKM	phosphofructokinase	112394271	1.09
CS	citrate synthase	112394084	1.07
IDH1	isocitrate dehydrogenase 1	112395270	1.42
IDH3A	isocitrate dehydrogenase 3 (NAD(+)) alpha	112405774	1.47
ACO1	aconitase 1	112405425	1.13
GLUD1	glutamate dehydrogenase 1	112390428	1.45
GLS	glutaminase	112392138	2.02
GLS2	glutaminase 2	112394275	1.59
INPP5A	inositol polyphosphate-5-phosphatase A	112411270	1.32
INPP4A	inositol polyphosphate-4-phosphatase type I A	112398954	1.62
IMPA1	inositol monophosphatase 1	112390609	1.26
NDUFA1	NADH:ubiquinone oxidoreductase subunit A1	112404044	−1.18
NDUFA6	NADH:ubiquinone oxidoreductase subunit A6	112394690	−1.69
NDUFS5	NADH:ubiquinone oxidoreductase subunit S5	112395486	−1.15
ATP5PD	ATP synthase peripheral stalk subunit d	112410517	−1.06
MIOX	myo-inositol oxygenase	112394676	−1.25
auditory-related differentially expressed genes			
MYO6	myosin VI	112390806	1.77
MYO7A	myosin VIIA	112402650	1.07
OTOF	otoferlin	112412779	2.69
OTOA	otoancorin	112399369	1.53
TECTB	tectorin beta	112406974	1.72
TRPV4	transient receptor potential cation channel subfamily V member 4	112397312	1.77
GNAO1	G protein subunit alpha o1	112403775	1.76
GRM2	glutamate metabotropic receptor 2	112400596	1.76
IQCE	IQ motif containing E	112412075	1.12
SLC38A1	solute carrier family 38 member 1	112394242	1.86
SLC38A2	solute carrier family 38 member 2	112394243	2.00
PLCB1	phospholipase C beta 1	112412530	1.68
PRKCB	protein kinase C beta	112399411	1.19
PLD1	phospholipase D1	112392372	1.54
GABRR2	gamma-aminobutyric acid type A receptor rho2 subunit	112401210	1.16
GABRB1	gamma-aminobutyric acid type A receptor beta1 subunit	112390625	2.92
GABBR2	gamma-aminobutyric acid type B receptor subunit 2	112392619	1.45
CACNA1F	calcium voltage-gated channel subunit alpha1 F	112399841	−2.18
HOMER3	homer scaffold protein 3	112414903	−1.06

## Data Availability

Data are contained within the article and Appendix A.

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
