# Peer review of "Comparative Blood Transcriptome Analysis of Semi-Natural and Controlled Environment Populations of Yangtze Finless Porpoise"

_animals, 2024, doi:10.3390/ani14020199_

Round 1
Reviewer 1 Report
Comments and Suggestions for Authors
I believe that it is an interesting work and that it achieves transcriptomics results in relation the differential expression between to group of population of the Yangtze finless porpoise in two different environments.
My major concern refers to the experimental design in which the two group of samples were collected at different times of the year and them to access and compare the differential gene expression between them. In fact, the different time collecting may impose different physiological requirements again environments. My question is how to circumvent this possibility and disentangling both factors different environments vs. different month of the year (space vs. time)?
Nevertheless, the authors have recovered the transcriptomic profile of DEGs in a good Discussion about the importance of different set of genes that could be explaining the different environment adaptation, and that could be followed in further studies, taking into account the disentangling both factors space vs. time.
I have only suggested small changes in particular in the Conclusion item.

Reviewer 2 Report
Comments and Suggestions for Authors
Dear authors,
This is a well written paper, but some issues that should be addressed before publishing. I believe the authors should not only cite the Cao et al. 2023 paper (now published in Fishes) but also discuss how the results presented in this study may relate to the findings in Cao et al. 2023 as they both seem to be asking similar questions (or it should be made more clear if question is different). Given this research, I think it would be relevant to look at the expression levels of genes that were found in Cao et al. 2023 and given this study design (animals sampled in different environments), how do the authors attempt to explain the differences in the conclusions?
In terms of the methodology, I think it is important to discuss more about the individual porpoises that were chosen to include in this study. What about the sex of the animals or age of the animals in this study and how might that have affected the outcome? Do the authors feel three individuals are adequate to draw conclusions? Can they please state something in the paper why so?
Cheers,
Comments on the Quality of English LanguageNo issues noted.
Author Response
Thanks for your valuable suggestions on our work. We have cited the paper "Blood Transcriptome Analysis Provides Responsive Changes in Gene Expression between Ex Situ and Captive Yangtze Finless Porpoises (Neophocaena asiaeorientalis asiaeorientalis) in the present MS. Our research team conducted a similar study at the same time. Cao et al. performed RNA sequencing of blood tissues from Yangtze finless porpoises in ex-situ and controlled environments, focusing on the impacts of environmental changes on genes related to vision, digestion, and immune system. It was found that the ex-situ population may enhance their immune function to adapt to the complex semi-natural environment, while the visual and protein digestion functions of the population in controlled environment were improved. In contrast, this study mainly focuses on the expression of metabolic and hearing-related genes between an ex-situ population and a controlled population of the Yangtze finless porpoise.
As a critically endangered species, the Yangtze finless porpoise is classified as national first-class key protected aquatic wildlife. It is difficult to obtain fresh tissue samples for the study of responsive changes in the Yangtze finless porpoises under different environments, so we used non-invasive blood collected during health checks for gene level testing. In this study, we tried to minimise interference with the experimental animals while meeting the statistical requirements (n=3). In order to minimize the impact of the physiological condition of the Yangtze finless porpoise on the experimental results, we selected physically healthy adult Yangtze finless porpoises with similar ages for the study. We focused the differences between the two groups of Yangtze finless porpoises on their environments, highlighting the main purpose of our study, which is to explore the intergroup differences of Yangtze finless porpoises in different environments, rather than the intragroup differences caused by different genders or ages. By analyzing the gene expression profiles of the semi-natural and controlled populations of Yangtze finless porpoises, our study has preliminarily revealed the various effects of environmental factors on them, providing a theoretical basis for further improving the quality of their habitats.
In addition, we have added the biological information in the section “Animals and sample collection”: six blood samples were collected from healthy Yangtze finless porpoises, including three YFPs with an average age of 5 years old from the Anqing Xijiang Yangtze finless porpoise ex-situ conservation base (XJ_1♂, XJ_2♂, XJ_3♂) in March 2018 and three YFPs with an average age of 7 years old from the Zhuhai Chimelong Yangtze finless porpoise artificial breeding and science popularization education base (CA_1♂, CA_2♀, CA_3♂) in November 2018.
Reviewer 3 Report
Comments and Suggestions for Authors
Dear authors,
thanks for giving me the opportunity to review your work. I found it very interesting despite it reports a small perspective focusing on a few samples and individuals without mid/long-term monitoring that could have shown a better overview and contribution to this study.
Please, find below some points which could be improved.
Many thanks
Lines 41-54: Please, include the rank of the species within the IUCN Red List.
Lines 55-71: Please, try to describe better the sense of this period. The research illustrated doesn't seem to have a link to each other.
Line 65: Please, replace "captive" with "under human care" or "controlled environment". Please, correct it within the whole text (see lines: 66, 85, 269, etc.).
Line 78: Please, specify which other "tissues" you mean. For example, blubber is the easiest tissue to sample compared to blood.
Line 389: Please, refer to your study as a "preliminary" study due to the few samples analyzed which could not be representative. Another limit to stress is the sampling conducted only 1 time and not during a period.
Line 393: Please, explain better which are the conservation insights of your study.
